# Urolithins Modulate the Viability, Autophagy, Apoptosis, and Nephrin Turnover in Podocytes Exposed to High Glucose

**DOI:** 10.3390/cells11162471

**Published:** 2022-08-09

**Authors:** Milena Kotewicz, Mirosława Krauze-Baranowska, Agnieszka Daca, Agata Płoska, Sylwia Godlewska, Leszek Kalinowski, Barbara Lewko

**Affiliations:** 1Department of Pharmaceutical Pathophysiology, Faculty of Pharmacy, Medical University of Gdansk, 80-210 Gdansk, Poland; 2Department of Pharmacognosy, Faculty of Pharmacy, Medical University of Gdansk, 80-210 Gdansk, Poland; 3Department of Pathology and Experimental Rheumatology, Faculty of Medicine, Medical University of Gdansk, 80-210 Gdansk, Poland; 4Department of Medical Laboratory Diagnostics-Fahrenheit Biobank BBMRI, Faculty of Pharmacy, Medical University of Gdansk, 80-210 Gdansk, Poland; 5BioTechMed Centre, Department of Mechanics of Materials and Structures, Gdansk University of Technology, 80-233 Gdansk, Poland

**Keywords:** podocytes, urolithins, urolithin A, high glucose, nephrin, diabetic nephropathy

## Abstract

Urolithins are bioactive compounds generated in human and animal intestines because of the bacterial metabolism of dietary ellagitannins (and their constituent, ellagic acid). Due to their multidirectional effects, including anti-inflammatory, antioxidant, anti-cancer, neuroprotective, and antiglycative properties, urolithins are potential novel therapeutic agents. In this study, while considering the future possibility of using urolithins to improve podocyte function in diabetes, we assessed the results of exposing mouse podocytes cultured in normal (NG, 5.5 mM) and high (HG, 25 mM) glucose concentrations to urolithin A (UA) and urolithin B (UB). Podocytes metabolized UA to form glucuronides in a time-dependent manner; however, in HG conditions, the metabolism was lower than in NG conditions. In HG milieu, UA improved podocyte viability more efficiently than UB and reduced the reactive oxygen species level. Both types of urolithins showed cytotoxic activity at high (100 µM) concentration. The UA upregulated total and surface nephrin expression, which was paralleled by enhanced nephrin internalization. Regulation of nephrin turnover was independent of ambient glucose concentration. We conclude that UA affects podocytes in different metabolic and functional aspects. With respect to its pro-survival effects in HG-induced toxicity, UA could be considered as a potent therapeutic candidate against diabetic podocytopathy.

## 1. Introduction

The current global diabetes prevalence is estimated to be over 10.5% of the population and is rising dramatically [1]. Type 2 diabetes mellitus (T2DM) accounts for more than 90% of all diabetes cases worldwide. Diabetic kidney disease is the most common cause of end-stage kidney disease in T2DM patients [2]. The prominent role of podocytes in etic kidney disease pathogenesis and progression has been demonstrated in numerous studies [3,4,5]. Accordingly, there is a consensus that podocytes are a primary therapeutic target to prevent the development of diabetic nephropathy [6,7,8]. Studies on the mechanisms of podocyte injury continue to provide new targets for treatment. Simultaneously, numerous novel therapeutic approaches are being investigated in vivo and in vitro to protect podocytes from impairment and loss. In this context, polyphenolic compounds are gaining more and more attention due to their broad spectrum of biological activities and therapeutic effects.

Urolithins comprise a group of polyphenolic compounds that are produced by gut microbiota after consuming ellagitannins and ellagic acid, complex polyphenols present in foods such as pomegranate, berries, and nuts. Dietary (poly)phenols are poorly absorbed in the intestine and after they reach the distal part of the gastrointestinal tract, they are metabolized by residing microbiota into urolithins D (UD), C (UC), A (UA), and B (UB), which are then transported into circulation (Figure 1) [9,10]. Most urolithins undergo phase II metabolism to form conjugates, mainly glucuronides and sulfates [11]. Although conjugated forms of urolithins are found in higher concentrations, free aglycones (unconjugated urolithins) show much higher biological activity [12]. Based on the differences in composition of gut microbiota between populations, three urolithin phenotypes (metabotypes) have been described: metabotype A (UA and UA conjugate producers), metabotype B (UB, UA, and iso-UA producers), and metabotype 0 (non-producers) [13]. In humans, metabotype A is predominant [14]. The diverse biological properties of urolithins include antioxidant, anti-inflammatory, anti-cancer, neuroprotective, antiglycative and other activities [15,16] that are exerted by the modulation of signaling molecules, cell structure, and protein expression [17]. Moreover, following human trials [18], urolithins are considered safe, and UA has been recently approved by the US Food and Drug Administration (FDA) as generally recognized as safe (GRAS) for its use as an ingredient.

Several in vitro and in vivo studies have indicated beneficial effects of urolithins in hyperglycemia-induced impairments in cellular and organ functions. Recently, the advantageous effects of urolithins in diabetic and non-diabetic kidney diseases have been reported [17,19]. Nevertheless, although it seems obvious that during glomerular filtration of urolithin-containing blood podocytes will directly contact these compounds, the few published studies conducted on renal cells do not include podocytes so far. Considering the potential impact of urolithin-based therapies on podocytes, as well as the future possibility of using urolithins for the improvement of podocyte function in diabetes, our goal was to assess the results of exposing mouse podocytes to UA and UB in conditions mimicking diabetes.

## 2. Materials and Methods

### 2.1. Urolithins

Urolithin A (UA, 3,8-dihydroxy-6H-dibenzo[b,d]pyran-6-one) and urolithin B (UB, 3-hydroxy-6H-dibenzo[b,d]pyran-6-one) were synthesized in the Department of Organic Chemistry at the Medical University of Gdańsk, based on literature data [20], and were kindly provided by the Department of Pharmacognosy and Department of Organic Chemistry, Medical University of Gdańsk, Poland. Urolithin A (228.2 g/mol) and UB (212.2 g/mol) were dissolved in sterile dimethyl sulfoxide (DMSO, Merck, Darmstadt, Germany), and 10 mM stock solutions were stored at −80 °C.

### 2.2. Podocyte Culture and Treatment

Immortalized mouse podocytes (SVI clone, Cell Line Services, Eppelheim, Germany) were cultured as described previously [21]. Briefly, the cells were propagated at 33 °C in RPMI1640 medium (PAN-Biotech, Aidenbach, Germany) containing 11 mM glucose and supplemented with 10% heat-inactivated fetal bovine serum (FBS, EURx Molecular Biology Products, Gdańsk, Poland), 100 U/mL penicillin, 100 µg/mL streptomycin (PAN-Biotech, Germany), and 10 U/mL recombinant mouse interferon-γ (IFN-γ, PeproTech EC, London, UK). Differentiation was induced by shifting the temperature to 37 °C. Podocytes were grown without IFN-γ, in Dulbecco’s Modified Eagle Medium (DMEM) containing 5.5 mM glucose (PAN-Biotech, Germany) and 5% FBS for 7–10 d. Subsequently, the cells were divided into two groups, one group remaining in DMEM with normal glucose (NG, 5.5 mM), and the other one was switched to high glucose (HG, 25 mM), and the culture was continued for the next 7 d. Culture media were changed every 2 d. Experimental NG or HG media containing 0.5% FBS and urolithins were added for the last 24 h or 48 h. The final DMSO concentration did not exceed 0.1% (*v*/*v*).

### 2.3. Chromatographic Determination of Urolithin A and Identification of Its Metabolite in Podocyte Culture Media

The HPLC-hypergrade (LC/MS) Lichrosolv solvents, acetonitrile, and methanol were purchased from Sigma-Aldrich (Steinheim, Germany). Analytical-grade formic acid (89–91% purity) was purchased from Merck (Darmstadt, Germany). Purified water was obtained from Milli-Q^®^ system (Molsheim, France).

The determined compounds were extracted from the podocyte culture according to the method of Sala et al. [22] modified by us. A mixture of 500 µL of cell culture media and 500 µL of cold methanol was vortexed for 1 min. The mixture was then centrifuged at 14,000 rpm for 10 min at room temperature. The supernatant was subjected to chromatographic analysis.

The chromatographic analysis was carried out using a HPLC-DAD-MS/MS Shimadzu system (Shimadzu Corp., Kyoto, Japan) (detailed data are reported in Appendix A) equipped with a Kinetex C-18 column (Phenomenex, Torrance, CA, USA; 2.6 µm, 100 mm × 2.1 mm) maintained at 25 °C. A triple-quadrupole LCMS 8040 (Shimadzu Corp., Japan) mass spectrometer equipped with an electrospray ionization (ESI) interface, operated in positive mode and negative mode, was used for mass spectrometric (MS) analysis. Analysis was performed by scanning from *m*/*z* 200 to 800 using a multiple reaction monitoring (MRM) technique. The operating conditions for the MS analysis were: DL (Desolvation Line) temperature 250 °C, heat block temperature 400 °C, nebulizing gas flow 3 L/min, drying gas flow 15 L/min, collision-induced dissociation gas pressure 230 kPa, detector voltage 1.9 kV, and interface voltage 4.5 kV for (+) ESI and 3.5 kV for (−) ESI. Dwell time was 100 ms. Data acquisition was performed with the software LabSolutions version 5.89 (Copyright© 2022–2016 Shimadzu Corp., Japan). Samples (1 µL for MRM and 6 μL for scanning) were analyzed with a use of gradient elution (Appendix A) during a total run time of 30 min. The procedure used was based on a modification of García-Villalba et al.’s [23] method. Under these conditions, UA was eluted with tR 13.53 min and its unknown metabolite (UAM) with tR 8.07 min. Both compounds had similar UV spectra—237,270 sh, 279,296, 305,354 and 267 sh; 278; 293; 302; 348, respectively, for UA and UAM, which were consistent with the literature data on urolithins and their metabolites [22]. The UAM was identified as urolithin 3-/or 8-O-glucuronide based on the *m*/*z* values of molecular ions [M + H]^+^ at *m*/*z* 405 and [M − H]^−^ at *m*/*z* 403 and fragment ion [M + H-glucuronosyl]^+^ at *m*/*z* 229.

### 2.4. Targeted Analysis of Urolithin A and Its Metabolite by Means of an MRM-Based Approach

Urolithin A and its metabolite were detected by tandem mass spectrometry (MS/MS) using MRM in positive mode by monitoring the transitions from precursor ions to dominant product ions, namely *m*/*z* 229.1→*m*/*z* 157.0 (+) (CE-22.0), *m*/*z* 229.1→*m*/*z* 128.1(+) (CE-40.0), *m*/*z* 229.1→*m*/*z* 185.0(+) (CE-18.0) for both compounds. The most intense transition was used for the quantitation (quantifier transition). The method was validated according to the guidelines of the FDA [24], including linearity range, limit of quantitation, limit of detection, repeatability, and extraction recovery for NG and HG media, separately (Appendix A).

### 2.5. Cell Viability Assay

Cell viability was determined using MTT assay. Podocytes were plated at 8000 cells per well in a 96-well plate and cultured as described above. Experimental media were added for the last 24 h or 48 h. After the indicated treatment period, the experimental media were removed, 100 µL of 0.5 mg/mL MTT was added to each well and cells were again incubated for 4 h at 37 °C. To dissolve the formazan crystals formed in the viable cells, 150 µL DMSO—Isopropanol (1:1, *v*/*v*) were added to each well. Following shaking the plates 3 × 1 min, absorbance was read at 570 nm in the SPECTROstar Nano (BMG LABTECH, Ortenberg, Germany). Results were analyzed using SPECTROstar Nano Mars software.

### 2.6. Flow Cytometry Analysis of Nephrin

Podocytes cultured in 6-well plates (90,000 cells/well) were rinsed with phosphate-buffered saline (PBS), detached by Accutase (STEMCELL Technologies, Köln, Germany), and centrifuged for 7 min at 400× *g*. Subsequently, the cells were fixed with 4% paraformaldehyde at room temperature for 8 min and blocked with blocking solution (2% FBS, 2% bovine serum albumin, 0.2% fish gelatin, in PBS) for 60 min at room temperature. Finally, the cells were resuspended in cold FACS buffer (2% FBS in PBS), and aliquots of 3 × 10^3^ cells/tube were incubated with phycoerythrin-conjugated antibody directed against the extracellular domain of nephrin (G-8, Santa Cruz) for 30 min at 4 °C. To omit debris and cell clumps, gating was performed, and 10^4^ gated events were counted. Cell fluorescence was analyzed using a BD FACSVerse™ Flow Cytometer (BD Biosciences, San Jose, CA, USA) and FlowJo^TM^ Software v10.8.0 (Ashland, OR, USA). Background fluorescence, assessed with an IgG isotype control, was subtracted from the corresponding samples during analysis.

### 2.7. Detection of Apoptosis

An Alexa Fluor^®^ 488 Annexin V/Dead Cell Apoptosis Kit and CellEvent Caspase-3/7 Green Detection Reagent with *SYTOX**™* AADvanced™ Dead Cell Stain (Invitrogen, Thermo Fisher Scientific, Rockwell, NY, USA) were used to quantify apoptosis according to the manufacturer’s instructions. Briefly, podocytes plated at 90,000 cells/well in a 6-well plate were cultured and treated with urolithins as described above. The cells were harvested by Accutase and washed with cold FACS buffer. For Annexin V assay, podocytes were resuspended in binding buffer, 5 µL Alexa Fluor^®^ 488 Annexin V, together with 1 µL propidium iodide (PI) working solution, were added to each 100 µL of cell suspension and the cells were incubated for 15 min at room temperature. For caspase 3/7 assay, podocytes were suspended in FACS buffer and 1 µL of CellEvent Caspase-3/7 Green Detection Reagent was added to all samples, which were then incubated for 30 min. During the last 5 min of incubation, 1 µL of SYTOX AADvanced dead cell stain solution was added to each sample and flow cytometry analysis was performed.

### 2.8. Immunofluorescence Staining and Confocal Microscopy

Immunofluorescence studies were performed as described previously [25]. Briefly, podocytes seeded on round glass coverslips (Bionovo, Legnica, Poland) were cultured in NG and HG media as indicated. Following exposure to various treatments, the cells were fixed with 4% paraformaldehyde for 8 min at room temperature, permeabilized (0.3% Triton X-100 in PBS) for 3 min and blocked with blocking solution as described in Section 2.6. The permeabilization step was omitted to visualize surface-bound antibodies. A 60-min incubation with primary antibodies (Appendix A) was followed by a subsequent 30-min incubation with secondary antibodies (Appendix A). All antibodies were diluted in blocking solution. Non-specific staining was controlled by replacing primary antibodies with blocking solution alone, which was followed by incubation with secondary antibodies. The coverslips were mounted on microscope slides using Fluoroshield^TM^ with DAPI (Merck, Darmstadt, Germany). Images were captured with the Opera Phenix^®^ Plus High-Content Screening System (Perkin Elmer, Waltham, MA, USA) and analyzed with Harmony High-Content Imaging and Analysis Software 4.8 (Perkin Elmer, Waltham, MA, USA). The images were merged using the ImageJ software (Version 1.53r, National Institutes of Health, University of Wisconsin, Madison, WI, USA), downloaded from: imagej.nih.gov/ij/ (accessed on 29 June 2022).

### 2.9. Assessment of Morphology and Actin Cytoskeleton

Podocytes grown in the culture plates were washed twice with PBS, fixed with 2% paraformaldehyde for 20 min, washed with deionized water and then stained with 0.1% crystal violet for 30 min at 37 °C to assess cell morphology. In separate experiments, podocytes were grown on the glass coverslips and immunostaining for F-actin was performed after washing with PBS and fixing with 4% paraformaldehyde, using phalloidin tagged to Alexa Fluor 488 dye (MoBiTec, Göttingen, Germany). Coverslips were mounted on the microscope slides using Fluoroshield^TM^ with DAPI. The images were taken with an inverted fluorescence microscope Zeiss Axioscope A1 (Carl Zeiss AG, Oberkochen, Germany), using NIS-Elements BR software (Precoptic, Warszawa, Poland).

### 2.10. RNA Isolation and qPCR

Total RNA from treated podocytes was extracted and purified with PureLink^TM^ RNA Mini Kit (Invitrogen, Carlsbad, CA, USA) according to the manufacturer’s instructions. The purity and integrity of the extracted RNA was checked with Cytation 3 multimode microplate reader (BioTek, Santa Clara, CA, USA) and analyzed by BioTek Gen5^TM^ 2.0 Data Analysis Software (Santa Clara, CA, USA). Quantitative polymerase chain reaction (qPCR) was performed using TaqMan RNA-to-C_T_^TM^ 1-step KIT (Applied Biosystems, Thermo Fisher Scientific), according to the manufacturer’s protocol. Briefly, the real-time (RT)-PCR reaction mix (TaqMan RT-PCR Mix, TaqMan RT Enzyme Mix, water) was combined with TaqMan^TM^ Gene Expression Assay for *Nphs1* gene encoding nephrin (Assay ID: Mm01176615_g1) or *Actb* gene encoding β-actin (Assay ID: Mm04394036_g1). A sample of 50 ng total RNA from each experimental group was added to 7.5 µL of Master Mix (10 µL total volume). The PCR reactions were carried out using QuantStudio 3 Real-Time PCR System (Applied Biosystems, Waltham, MA, USA) and involved the following steps: (1) reverse transcription at 48 °C for 20 min; (2) polymerase activation at 95 °C for 10 min; (3) 40 cycles denaturation (15 s at 95 °C) followed by annealing/extending at 60 °C for 1 min. Relative levels of target gene mRNA expression were normalized to β-actin and the relative level of mRNA was calculated with the ΔΔ comparative threshold (Ct) method.

### 2.11. Protein Extraction and Western Blot Analysis

Podocytes subjected to the different experimental conditions were lysed using Pierce^TM^ RIPA Buffer (Thermo Fisher Scientific), containing Halt^TM^ Protease and Phosphatase Single-Use Inhibitor Cocktail (Thermo Fisher Scientific), and proteins were extracted from the cells according to the manufacturer’s protocol. Total protein concentration was determined by DC Protein Assay (Bio-Rad Laboratories, Hercules, CA, USA). Proteins (20–30 µg) were separated by Criterion^TM^ TGX Stain-Free^TM^ Precast Gel electrophoresis (Bio-Rad Laboratories, Hercules, CA, USA) and transferred to polyvinylidene difluoride (PVDF) membrane (Trans-Blot Turbo, Midi Format, 0.2 µm PVDF) using Trans-Blot Turbo Transfer system (Bio-Rad Laboratories, USA). Next, the membranes were blocked with 5% bovine serum albumin in TBST buffer (Tris Buffered Saline with Tween 20) for 30 min and incubated overnight with primary antibodies (Appendix A). Sodium Potassium ATPase (SPA) or β-actin expression was analyzed to ensure equal protein loading. The membranes were washed in TBST and incubated with horseradish peroxidase–linked secondary antibody (Appendix A). The proteins were then visualized by VisiGlo^TM^ Select HRP Substrate Kit (VWR Chemicals, Aurora, OH, USA) and imaged using a ChemiDoc MP (Bio-Rad Laboratories, Hercules, CA, USA). Densitometry was performed using ImageLab v2.0 analysis software (Bio-Rad Laboratories, Hercules, CA, USA).

### 2.12. Statistical Analyses

All data are shown as mean ± SEM and were compared by ANOVA, Mann–Whitney, or the Student’s *t*-test to test for statistical significance. *p* values < 0.05 were considered statistically significant.

## 3. Results

### 3.1. Urolithin A Is Metabolized by Podocytes

In in vivo conditions, the majority of urolithins absorbed from the gut undergo phase II metabolism, forming conjugates, of which glucuronides are the most abundant [9]. Additionally, in the cell cultures, final free urolithin concentration can be modified by metabolic processes [26]. Peripheral metabolism of urolithins by podocytes is a key point to be considered during research. Therefore, we examined the ability of podocytes to metabolize UA in NG and HG conditions. The stability of UA, as well as the appearance of its newly formed metabolites, was studied in NG and HG cell media containing 10 µM UA after 12 h, 24 h, and 48 h of incubation. A sensitive HPLC-MS/MS analysis was performed to target the UA concentration and to identify different UA metabolites that could be formed under the experimental conditions.

The UA molecules were stable under the experimental conditions (cell-free incubation), in contrast to collected cell culture supernatants. The analysis of the concentration of UA in the medium over time, both with NG and HG concentrations, showed a decrease in the concentration of the parent molecule in favor of the emerging metabolite. The 3-O/ or 8-O-glucuronide UA (UAM), the metabolite of UA in the cell culture medium, was detected after 12 h of incubation and increased as the time of incubation passed (Figure 2A). The time-course analysis of UA revealed statistically significant differences of aglycone concentrations in NG. The % of aglycone was significantly reduced by 35% after 24 h (*p* < 0.01) and by 59% after 48 h (*p* < 0.003) as compared to the 12-h group (Figure 2B). There was no significant difference in metabolic capacity between the NG and HG groups, but metabolism of urolithins by podocytes seemed to be lower in HG conditions (Figure 2C).

Based on these results, for 48-h incubation periods, incubation media were replaced with fresh urolithin solutions after 24 h.

### 3.2. Podocyte Viability and Structure Are Affected by High Concentrations of Urolithins

Dependent on the cell type, a broad range of urolithin concentrations has been reported to be effective in in vitro experiments [27,28,29]. Thus, to establish conditions for our further experiments, we first tested the viability of NG podocytes that were exposed to increasing concentrations of UA and UB for 24 h and 48 h. Following the 24-h treatment, MTT assay showed a significant decline in the viability of podocytes incubated with 10 µM UB, whereas a pronounced drop in the presence of both UA and UB was observed at a 100 µM concentration (Figure 3A). Results of prolonged, 48-h incubation with 10, 30, and 100 µM UA and UB revealed that only 10 µM UA did not affect podocyte viability, while at higher concentrations of both urolithins, the viability gradually declined (Figure 3B).

Moreover, in UB-treated podocytes, the percentage of viable cells significantly dropped at 10 µM UB concentration (*p* < 0.01 vs. Control). On the other hand, in podocytes cultured in HG, UA and UB affected the cell viability in a different manner (Figure 4). HG alone significantly decreased the percentage of viable podocytes (*p* < 0.01 vs. NG Control), whereas in the cells treated for 48 h with 10 µM UA, the viability was improved, as compared to HG Control (*p* < 0.005). Nevertheless, it was still below the value for untreated NG cells (*p* < 0.05). An adverse effect was observed in 10 µM UB–treated podocytes. In this group, the percentage of viable cells was markedly lower than in the respective UA group (*p* < 0.005). Increasing urolithin concentrations resulted in further decline in the number of viable cells. Yet, at all concentrations, the percentage of viable UA-treated podocytes was slightly higher than in UB-treated cells.

The function of podocytes is strictly correlated with the maintenance of the actin cytoskeleton [30]. Remodeling of actin fiber organization may reflect cell damage triggered by biological and chemical factors. Therefore, we investigated the impact of urolithin concentration on podocyte morphology and cytoskeleton. Crystal violet and F-actin staining clearly demonstrated that there were no apparent differences between the architecture of control podocytes and the cells exposed for 24 h to 10 µM UA and UB. High glucose itself rearranged F-actin fibers to form thick cortical bundles, and urolithins apparently did not seem to modulate this effect. However, at 100 µM urolithins, profound changes occurred, such as cell body narrowing, shrinking, and rearrangement of F-actin fibers (Figure 5). The results confirmed that exposure to 100 µM urolithins is deleterious to podocytes.

Urolithin-dependent changes in podocyte viability could be due to regulation by urolithins of apoptotic and/or necrotic mechanisms. The Bcl-2 protein family members play a pivotal role in regulating cell apoptosis [31]. Thus, using Western blot analysis, we first examined whether Bcl-2 expression was modulated by UA and UB. Furthermore, based on the MTT results, our intention was to check whether Bcl-2 expression was differentially regulated by UA and UB in HG-treated cells. In general, Bcl-2 upregulation is associated with anti-apoptotic mechanisms [32]. Therefore, we were surprised to find that in both the NG and HG groups, despite loss of viability, the Bcl-2 expression was markedly higher at 100 µM UA and UB than in the control cells (Figure 6). On the other hand, pronounced downregulation of Bcl-2 in the control cells cultured for 7 d in HG (*p* < 0.001 vs. NG Control) suggested that HG induced podocyte apoptosis, which was in accordance with numerous other studies [33,34]. Moreover, both 10 µM UA and 10 µM UB elevated the Bcl-2 level in the NG group. Conversely, in the HG group only, 10 µM UA upregulated Bcl-2 expression.

Next, we cytometrically quantified apoptosis in podocytes incubated with UA. Results of Annexin V/PI double-staining disclosed that in the NG group, the percentage of early and late apoptotic cells significantly increased at 100 µM UA (Figure 7B), which was most likely the reason for the previously observed decline in podocyte viability. Exposure of podocytes to HG induced early and late apoptosis (Figure 7C), which was consistent with earlier MTT and Bcl-2 results. The percentage of early apoptotic cells was markedly reduced at 10 µM UA (*p* < 0.05 vs. Control) while a considerable increase occurred at 100 µM UA (*p* < 0.01 vs. 10 µM UA). These results indicate that apoptosis induced by prolonged treatment of podocytes with HG can be at least partially reversed by 10 µM UA.

We also measured caspase 3/7 activity to confirm that the reduced podocyte viability reflected changes in podocyte apoptosis. The cells were simultaneously stained with SYTOX to assess the percentage of dead cells. The results from flow cytometry revealed that in both NG and HG cells, caspase 3/7 was activated upon treatment of podocytes with 100 µM UA (Figure 7D). Alone, HG significantly increased caspase activity compared with that in the NG group (Figure 7F, *p* < 0.005). Interestingly, compared to the control, 10 µM UA reduced the activity of caspase 3/7 not only in HG (*p* < 0.05), but also in NG-treated podocytes (*p* < 0.01). Taken together, these results suggested that in podocytes exposed to HG, 10 µM UA treatment may improve cell viability by decreasing the rate of apoptosis. Conversely, 100 µM UA triggered podocyte apoptosis, which was aggravated by the HG milieu.

Based on the above results, in our further experiments, we applied 10 µM urolithin concentration and 24 h incubation time, unless indicated otherwise.

### 3.3. Urolithin A Reduces High Glucose–Induced Reactive Oxygen Species Production in Podocytes

Mechanisms involved in the HG-induced apoptosis include elevation of intracellular reactive oxygen species (ROS) [35]. On the other hand, similarly to other polyphenolic compounds, antioxidative properties of urolithins, including UA, have been well documented [36]. Hence, we examined whether the effects of UA and UB on podocyte viability observed by us could be associated with the regulation of ROS levels by urolithins. Flow cytometric analysis confirmed that in HG-treated podocytes, ROS levels increased almost fivefold, as compared to the NG cells (Figure 8, *p* < 0.01). In podocytes incubated with 10 µM UA, ROS production was prominently downregulated (*p* < 0.05 vs. HG Control), whereas there was no significant effect of incubation with 10 µM UB. That UB was a much less potent antioxidant than UA was consistent with previous observations [20]. As oxidative stress plays a pivotal role in apoptosis [37], our results suggest that anti-apoptotic effects of UA in podocytes exposed to HG are associated with the reduction of ROS levels, which most likely accounts for its antioxidant activity.

### 3.4. Urolithin A Modulates Expression of Autophagy-Related Proteins

Cell viability is regulated by an interplay between apoptosis and autophagy [38]. Opposite to apoptosis, autophagy generally enhances cell survival; however, in some circumstances it also may lead to cell death [39]. It has been demonstrated in different studies that HG either impairs [7,40,41] or stimulates [42,43] autophagy in podocytes, and the results seem to depend on duration of cell exposure to the hyperglycemic milieu. On the other hand, urolithins have been shown to regulate autophagy in some cell types [44,45]. To ascertain if UA-dependent effects on podocyte viability and apoptosis were associated with the modulation of autophagy, we next examined the expression of three commonly used autophagy markers, light chain 3B protein (LC3B), autophagy-related 5 protein (ATG5), and p62. The LC3B is bound to autophagosomes, while ATG5 is a key regulator of autophagy, and the levels of both proteins are directly correlated with autophagic flux [46]. In examining LC3B levels on Western blot analysis, we observed no significant changes in LC3B expression in HG-treated cells (Figure 9B). However, as compared to the NG Control, the ATG5 levels were markedly downregulated (*p* < 0.001, Figure 9C), suggesting that HG impaired autophagic flux in podocytes. Upon treatment with 10 µM UA, LC3B and ATG5 levels increased significantly in HG cells (*p* < 0.01 vs. NG Control), and LC3B was also upregulated by UA in the NG group (*p* < 0.001 vs. NG Control). These results indicate that autophagy in podocytes was induced in response to UA. Accordingly, pronounced differences between immunofluorescent staining in the control and UA-treated podocytes were observed by confocal microscopy (Figure 9E). Unexpectedly, p62 protein, which is degraded during autophagy, was upregulated by UA in both NG and HG cells (Figure 9D, *p* < 0.001 vs. respective Control), which seemed to be in contrast with the LCB3 and ATG5 results. Quantitative analysis of confocal images confirmed that UA increased the expression of p62 (Figure 9F). Simultaneously, upregulated p62 levels in non-treated HG cells suggested that autophagy was reduced by HG, which was in concert with ATG5 Western blot analysis.

### 3.5. Urolithin A Upregulates Nephrin Protein but Reduces mRNA Expression

Hyperglycemia-induced impairment of nephrin expression is associated with severe podocyte dysfunction. It has been previously reported that some polyphenolic compounds modulated the expression of nephrin in vivo and in vitro [47,48]. Hence, we next examined whether UA affected the expression of nephrin in podocytes cultured in the NG and HG media. Considering that urolithins can regulate gene expression [27,49], we first performed the quantitative RT-PCR analysis of *NPHS1* gene in podocytes treated with 10 µM UA. Results revealed that in podocytes from the HG group, nephrin mRNA was significantly decreased (*p* < 0.01, Figure 10). Moreover, 24-h treatment of podocytes with UA significantly reduced the expression of *NPHS1* gene in podocytes from NG (*p* < 0.01), as well as from the HG group (*p* < 0.05).

To determine if UA-induced changes in nephrin expression were mirrored by respective changes in protein levels, we performed a Western blot analysis that confirmed that total nephrin expression was significantly downregulated in podocytes exposed to HG (*p* < 0.05 vs. NG, Figure 11). Yet, in contrast to the qPCR results, upon treatment with 10 µM UA, nephrin expression was markedly upregulated in NG cells (*p* < 0.01 vs. NG Control) and was also elevated, however, not significantly, in HG-treated podocytes.

### 3.6. Urolithin A Upregulates Nephrin Expression at the Podocyte Surface

Proper nephrin expression at the podocyte surface is essential for podocyte function, so we focused our next analysis on nephrin localized at the plasma membrane. Using an antibody directed to the extracellular nephrin domain, we performed a flow cytometry assay of podocytes treated as above. The results were in line with these from the Western blot analysis and showed not only the HG-induced drop in surface-bound nephrin (Figure 12A), but also a pronounced nephrin upregulation in response to 10 µM UA in both NG and HG podocytes (*p* < 0.001).

### 3.7. Intracellular Trafficking of Nephrin Is Modulated by Urolithin A

We next attempted to explain the striking discrepancy between the effects of UA on nephrin mRNA and protein expression. One of the possibilities was that while simultaneously suppressing the *NPHS1* gene, UA suppressed the rate of nephrin endocytosis and slowed down nephrin turnover. To check this hypothesis, we quantified the co-localization of nephrin and early endosome marker EEA1 (Figure 13). However, confocal analysis of double-stained cells disclosed that upon UA treatment, the extent of intracellular co-localization of nephrin end EEA1 markedly increased in both NG and HG cells (*p* < 0.001). These results indicate that UA enhanced nephrin trafficking and the effect was independent on glucose concentration.

## 4. Discussion

Urolithins, the products of gut microbiota from ellagitannin rich foodstuff were identified in humans almost 20 years ago [50]. Since then, numerous in vitro and in vivo studies have demonstrated that endogenous as well as exogenously delivered urolithins are potent multifunctional compounds capable of regulating a variety of cellular processes [11]. Due to their immense health benefits, urolithins are under consideration for the application in the treatment of several diseases, including diabetes [51]. Some urolithins circulating in plasma accumulate in certain tissues, whereas the remaining urolithins pass through the glomerular filtration barrier, directly contacting its components. Nevertheless, almost no publications address the action of these compounds on podocytes. Following the establishment of a HG-induced podocyte injury model, in our present study, we demonstrated that urolithins affected podocyte viability and nephrin expression, as well as nephrin endosomal trafficking. We revealed that in HG-treated podocytes, UA decreased the rate of apoptosis, upregulated autophagic flux, and inhibited HG-induced ROS production. Our further finding was that independent of ambient glucose concentration, UA upregulated nephrin expression at the podocyte surface and, most likely, accelerated nephrin turnover. Moreover, we showed that podocytes metabolized urolithins by glucuronidation.

Urolithins are highly lipophilic and thus can easily cross cell membranes. After being produced in the large intestine, urolithins undergo phase II metabolism (mostly glucuronidation) in intestinal enterocytes and hepatocytes [10,23]. In our experiments we have demonstrated that podocytes also actively converted free aglycones to glucuronides that were then released from the cells into the culture media (Figure 2). In podocytes cultured in NG, the rate of glucuronidation was relatively high (69% of free aglycone conjugated after 48-h incubation with UA). However, podocytes from the HG group exhibited slower metabolism of UA than NG cells, resulting in a 48% decrease of UA after 48 h. Compared to UA conjugates, free aglycones show much higher biological activity [13,52,53]. Experiments in rats revealed that systemic inflammation triggered tissue deconjugation of UA-glucuronide to free UA [52], which was probably a mechanism enhancing the anti-inflammatory activity of urolithin. It is well established that HG induces oxidative stress and activates a number of inflammatory pathways [47,54]. In our experiments, UA profoundly decreased ROS production in podocytes from the HG group (Figure 8). Hence, we hypothesize that slowing down the glucuronidation rate of UA in podocytes exposed to HG could be protective against the deleterious effects of HG.

Different dose-dependent effects of urolithins on cell viability in vitro were observed in various cell types [29,44,55,56]. In some cells, UA and UB did not show any cytotoxic effects up to the concentration of 100 µM [27]; therefore, we tested physiological (10–30 µM) as well as high (100 µM) concentrations of UA and UB. However, we found that in NG conditions, only 10 µM UA was not toxic, whereas UB already at that low concentration impaired podocyte viability (Figure 3). Consistent with previous reports [57,58], exposure to HG reduced podocyte viability (Figure 4) and increased apoptotic rate (Figure 7A–C), a well as caspase 3/7 activity (Figure 7D–F), with concomitant downregulation of anti-apoptotic protein Bcl-2 (Figure 6). Treatment of podocytes with 10 µM UA suppressed apoptosis and caspase activity and improved cell viability, while UB at all tested concentrations aggravated the toxic effect of HG (Figure 4). Yet, at 100 µM concentrations, both UA and UB dramatically reduced the viability of podocytes from the NG and HG groups (Figure 3 and Figure 4), which was accompanied by the disruption of the cell architecture (Figure 5) and increased apoptotic rate (Figure 7). Nevertheless, at the same time, the anti-apoptotic protein Bcl-2 was upregulated, which could be a defensive mechanism toward apoptotic stimuli, as has been proposed for some other cells [59,60]. The Bcl-2 protein is also involved in the inhibition of autophagy, another catabolic process essential for cellular homeostasis. The predominant function of autophagy is to promote cell survival, although under some circumstances it is linked to cell death [61]. Together with ATG5, Bcl-2 is responsible for crosstalk among apoptosis and autophagy [62]. In the context of our results, Bcl-2 upregulation at 100 µM UA could be directed against UA toxicity, as a cytoprotective mechanism abolishing the cell death-inducing autophagy. On the other hand, subcellular localization of Bcl-2 is distributed between mitochondria—which are the main site of Bcl-2 expression—and other organelles. A small portion is localized to the endoplasmic reticulum (ER) membrane, and only ER-targeted Bcl-2, but not mitochondrial Bcl-2, inhibits autophagy [63,64]. However, in this study, we did not investigate the subcellular expression of Bcl-2 in podocytes exposed to UA.

In different in vitro studies on podocytes, HG has been reported either to induce [42] or to attenuate [41] autophagy. This apparent discrepancy has been cleared up in a study in which the authors demonstrated that the autophagic flux depended on the duration of exposure to HG. Autophagy was induced after a short-term (48-h) treatment of podocytes with HG, while it was repressed after a long-term (15-d) exposure to HG [43]. In line with this finding, our study revealed that the long-term (7-d) exposure of podocytes to HG suppressed autophagy, which was reflected by downregulation of ATG5 and upregulation of p62 levels (Figure 9C,F). At the same time, the level of autophagosome marker LC3B remained unchanged (Figure 9B). Yet, the number of autophagosomes does not always correspond to the autophagic activity. As a matter of fact, it is resultant of the balance between the rate of their generation and the rate of their conversion into autolysosomes. Thus, lack of LC3B downregulation may represent suppression of autophagy downstream of autophagosome formation [65]. However, upon treatment with 10 µM UA, both LC3B and ATG5 were upregulated, which indicates that UA stimulated autophagy in HG, and most likely in NG podocytes. These results suggest that in HG-treated podocytes, UA could induce recovery from an apoptotic state by upregulating autophagic flux.

As it is recommended that the measurement of autophagic flux be performed in combination with several markers, we also determined the expression of p62, also known as SQSTM1/sequestome 1. The p62 molecules are selectively incorporated into autophagosomes through direct binding to LC3 and are degraded by autophagy. Thus, the total cellular expression levels of p62 inversely correlates with autophagic activity, and p62 is frequently used to monitor autophagic flux [66]. Therefore, we were astonished to realize that in podocytes treated with 10 µM UA, p62 was significantly upregulated (Figure 9D–F), which is usually associated with the reduction of autophagic activity. Yet, several factors, including transcriptional upregulation, can affect the whole-cell p62 level under certain conditions [66,67]. The transcriptional activity of urolithins has been documented in several studies [44,68,69], which makes it likely that UA could affect the expression of p62 independent of autophagy. Furthermore, in addition to its cytosolic functions, including involvement in autophagy, p62 also forms nuclear bodies, the role of which is not fully understood [70].

Normal nephrin expression is critical for slit diaphragm integrity and podocyte structure and function. Disturbances in nephrin abundance and cellular localization are considered to cause podocyte injury and loss [71,72], which is in turn associated with the majority of glomerular diseases [73]. Several mechanisms have been postulated to mediate podocyte depletion in nephrin-deficient glomeruli. Surprisingly, disturbances in nephrin expression are not associated with apoptosis [74]. Proteinuria, a hallmark of many glomerular diseases, precedes podocyte impairment. Among the few published studies on the effects of urolithins on kidney function, one study showed UB ameliorated proteinuria in a unilateral ureteral obstruction rat model [75]. Thus, exploring the effects of UA on nephrin expression seemed to be particularly interesting to us.

High glucose downregulated nephrin at both mRNA (Figure 10) and total protein (Figure 11) levels, which was consistent with previously published data [76,77,78]. Simultaneously, HG-treated podocytes expressed less nephrin at the surface (Figure 12). In the presence of UA, surface nephrin was increased, which was accompanied by the elevation of total nephrin expression in the whole-cell lysates. However, this was accompanied by a pronounced drop of NPHS1 mRNA levels, which was an unexpected finding. Discrepancy between the abundance of cognate protein and RNA molecules is frequently observed, whereas quantitative relations between RNA and protein are still not fully understood [79,80]. It has been proposed that such an inverse correlation may be due to the regulation of translation [81] and is frequently associated with the molecular and structural polarity of some cells [82], such as podocytes. Urolithin A is known to modulate multiple intracellular signaling pathways [28], so we can speculate that UA also affected transcriptional activity in podocytes.

Some studies have revealed that the decreased abundance of surface nephrin was associated with enhanced internalization of the protein [83,84]. Hence, we investigated in a converse situation, when nephrin expression at the podocyte surface was upregulated by UA, whether nephrin endocytic trafficking could be impaired. Based on the detailed study on nephrin incorporation into endosomal structures [85], we examined the intracellular co-localization of nephrin with early endosome marker EAA1 that was shown to highly co-localize with nephrin at an early phase, as well as over a long time of internalization. In podocytes exposed to HG, nephrin incorporation into EAA1-positive compartments increased, which indicated that endocytosis was enhanced in HG cells, as described previously [83]. However, as shown in Figure 13B, in both NG- and HG-treated podocytes, incubation with UA further augmented nephrin trafficking. Thus, both synthesis and endocytosis of nephrin were augmented by UA, which suggests that UA accelerated nephrin turnover. Of all the urolithin isoforms, UA and UB are the most frequently tested compounds. Both urolithins have been shown to exert various in vitro and in vivo biological activities, that in general are considered to be beneficial [11,17]. In this study, we show that in HG-treated podocytes, UA demonstrated its cytoprotective properties more effectively than UB. Moreover, UB seemed to negatively affect podocyte viability more readily than UA. Thus, our results suggest that in HG milieu mimicking diabetes, UA may have higher therapeutic potential than UB. However, one should note that concentrations of highly bioactive free aglycones in the culture media were higher than in the in vivo conditions. In plasma, as well as in tissues, most urolithins are found in a less active conjugated form, no matter if administered orally or directly injected [9,86]. So far, various conditions (e.g., concentration, incubation time, conjugates, or free aglycones) for in vitro studies on urolithins have been used by different authors. It still remains questionable, however, to what extent the experiments yield physiologically relevant results [16]. Nevertheless, it is tempting to consider future treatment of podocytes (e.g., with nanoparticle carriers that would transport active urolithin aglycones directly to the target cells). In this context, in vitro studies may provide essential data on safety and efficiency of such treatment.

In summary, our results show that urolithins are involved in mechanisms regulating podocyte viability. Observed in HG milieu, the pro-survival effects of urolithins, and particularly those of UA, make these polyphenolic compounds potent therapeutic candidates against podocyte impairment in diabetes. Slowing down UA glucuronidation in HG milieu could represent the adjustment of podocytes to the current demand for highly active free aglycone. Furthermore, enhanced by UA, nephrin turnover suggests that slit diaphragm integrity could be modulated by this compound. However, considering their pleiotropic activities, urolithins need to be carefully studied with respect to potential long-term merits and the adverse effects.

## Figures and Tables

**Figure 1 cells-11-02471-f001:**
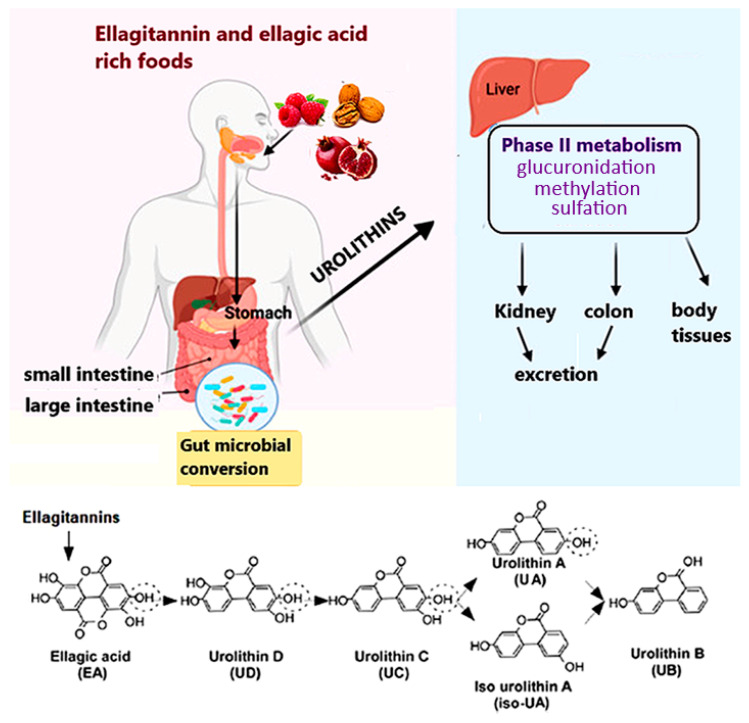
Ellagitannins and their hydrolysable derivative, ellagic acid, are natural polyphenols abundant in some fruits, nuts, and seeds. Ellagitannin is metabolized by gut microbiota into distinct types of urolithins that are further converted by large intestine enterocytes and the liver into conjugates. Both conjugated and unconjugated urolithins enter the blood circulation system and reach the body tissues. Besides accumulation in certain organs, urolithins are excreted in urine and feces [9]. Figure was adapted from [10,17].

**Figure 2 cells-11-02471-f002:**
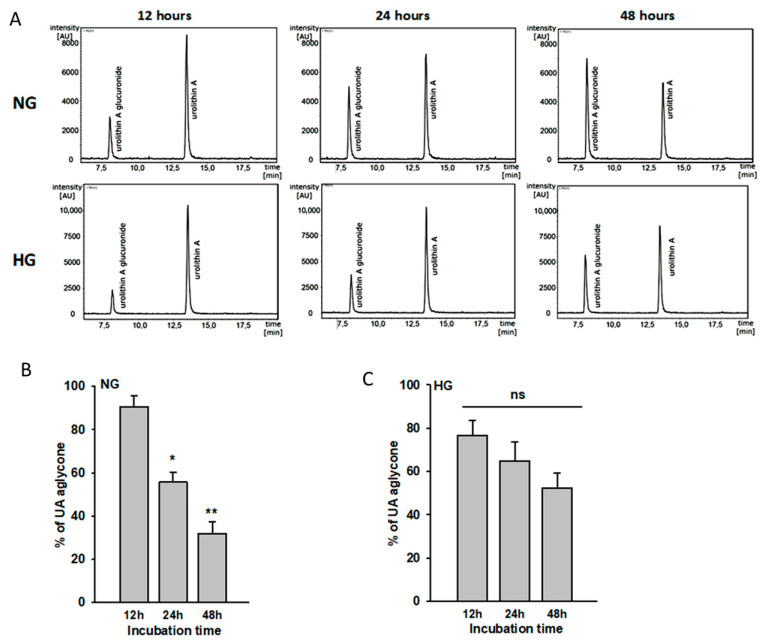
Urolithin A is metabolized by podocytes. Podocytes cultured for 7 d in normal (5.5 mM, NG) or high (25 mM, HG) glucose were incubated with 10 µM UA for the indicated time periods. At time 0, concentration of UA aglycone was 10 µM (100%). The presence of urolithin and its glucuronide in the culture medium was assessed using HPLC-DAD-MS/MS after 12 h, 24 h, and 48 h of incubation (**A**). Calculated results (**B**,**C**) show percentage of free aglycone detected in culture media and are expressed as mean ± SEM from three independent experiments performed in duplicate. The Student’s *t*-test was used to calculate *p* values. * *p* < 0.01 24 h vs. 12 h, ** *p* < 0.003 48 h vs. 12 h. AU: arbitrary units; ns: not significant.

**Figure 3 cells-11-02471-f003:**
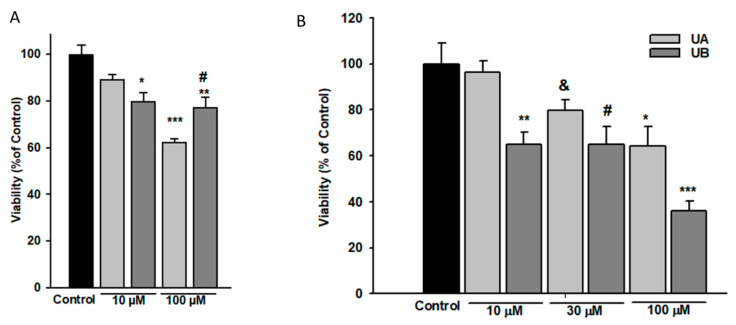
Podocyte viability was impaired by high concentrations of urolithins. Podocytes cultured in normal glucose (5.5 mM, NG) were incubated for 24 h (**A**) or 48 h (**B**) with varying concentrations of urolithins and analyzed by MTT test. Results show mean ± SEM. Control cells were incubated with vehicle (DMSO) alone. ANOVA test was used to calculate *p* values. For 24-h incubation (**A**) * *p* < 0.05 vs. Control, ** *p* < 0.01 vs. Control, *** *p* < 0.001 vs. Control and 10 µM UA, # *p* < 0.05 vs. 100 µM UA. For 48-h incubation (**B**) * *p* < 0.05 100 µM UA vs. 100 µM UB, vs. 10 µM UA, and vs. Control, # *p* < 0.05 vs. Control, & *p* < 0.05 vs. 10 µM UA, ** *p* < 0.005 vs. 10 µM UA and vs. Control, *** *p* < 0.001 vs. Control. Each MTT test was performed in quadruplicate (*n* = 4).

**Figure 4 cells-11-02471-f004:**
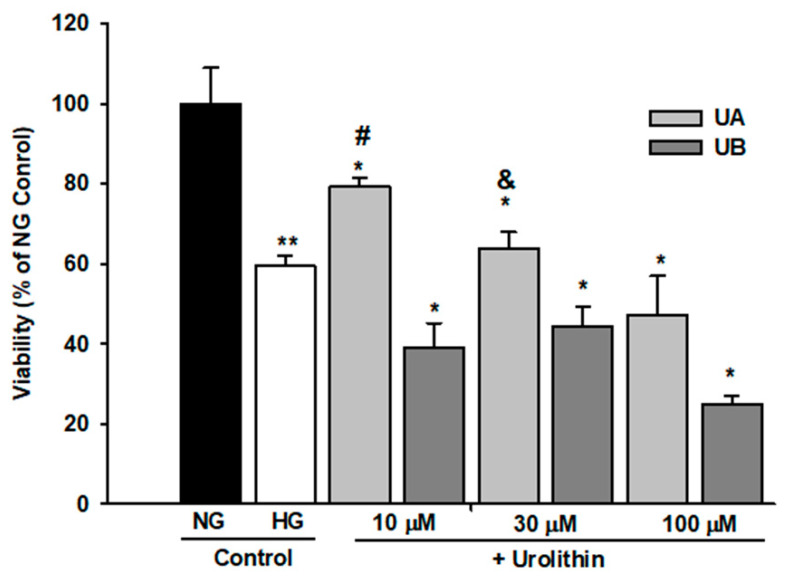
Urolithins A and B differentially affected the viability of podocytes exposed to high glucose. Podocytes cultured in high glucose (25 mM, HG) were incubated for 48 h with varying concentrations of urolithins. Control cells were incubated with vehicle (DMSO) alone. Results show mean ± SEM. ANOVA and the Student’s *t*-tests were used to calculate *p* values. * *p* < 0.05 vs. NG Control, & *p* < 0.05 vs. 30 µM UB, ** *p* < 0.01 vs. NG Control and vs. 100 µM UB, # *p* < 0.005 vs. HG Control and vs. 10 µM UB. Each MTT test was performed in quadruplicate (*n* = 4).

**Figure 5 cells-11-02471-f005:**
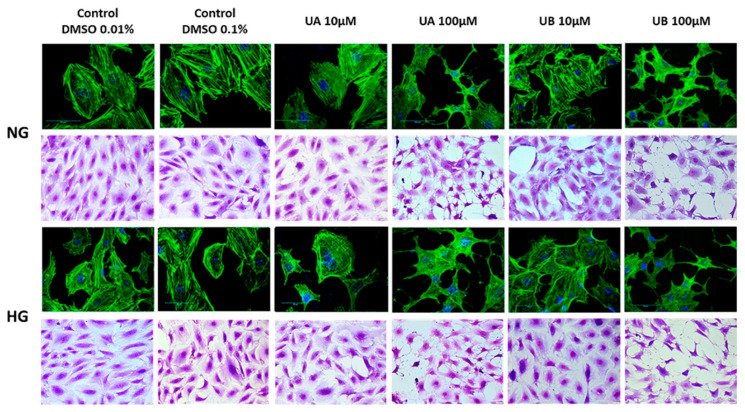
Urolithins at high concentrations disrupted podocyte morphology and structure. Podocytes were cultured on glass coverslips for 7 d in normal (5.5 mM, NG) or high (25 mM, HG) glucose. For the last 48 h, the cells were incubated with 10 µM and 100 µM UA or UB and subsequently stained with crystal violet or phalloidin/Alexa 488. Media containing respective vehicle concentrations served as the controls. Representative pictures out of three independent experiments.

**Figure 6 cells-11-02471-f006:**
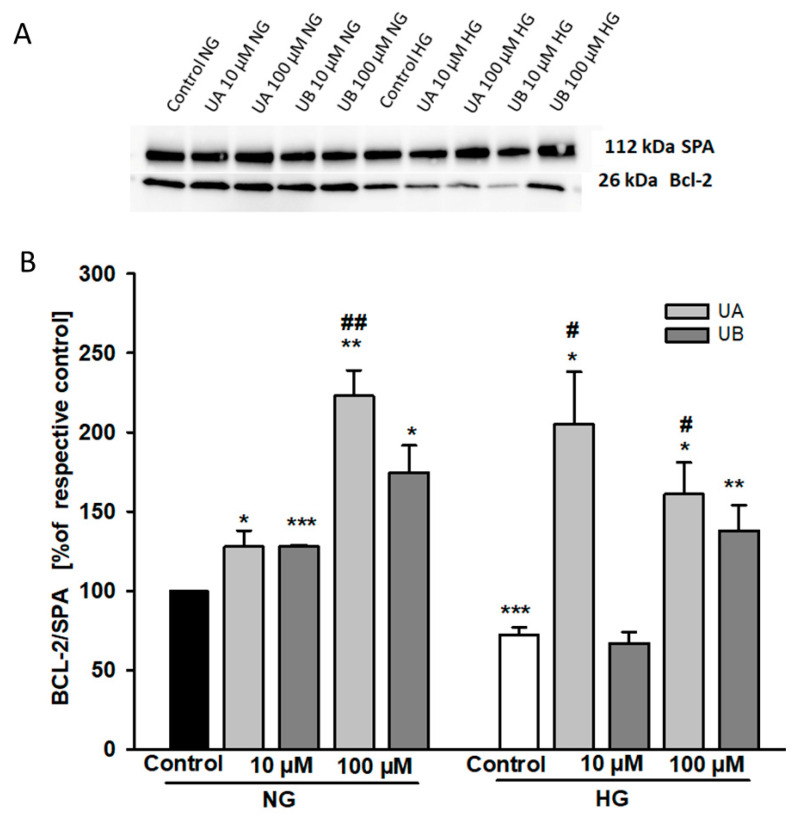
Urolithins and high glucose modulate the expression of Bcl-2. Podocytes cultured for 7 d in normal (5.5 mM, NG) or high (25 mM, HG) glucose were incubated for 24 h with 10 µM UA and 10 µM UB. 20-µg protein samples from total cell lysates were subjected to Western blot analysis followed by quantitative densitometric analysis. Representative immunoblot (**A**). Quantification results of the ratio of Bcl-2 to Alpha 1 Sodium Potassium ATPase (SPA) are shown as % of respective Control (**B**). HG Control is expressed as % of the NG Control. Results show mean ± SEM. ANOVA and the Student’s *t*-tests were used to calculate *p* values. * *p* < 0.05 vs. respective Control, ** *p* < 0.02 vs. respective Control, *** *p* < 0.001 vs. NG Control, # *p* < 0.05 vs. HG-UB10 mM, ## *p* < 0.02 vs. NG-10 µM UA (*n* = 4).

**Figure 7 cells-11-02471-f007:**
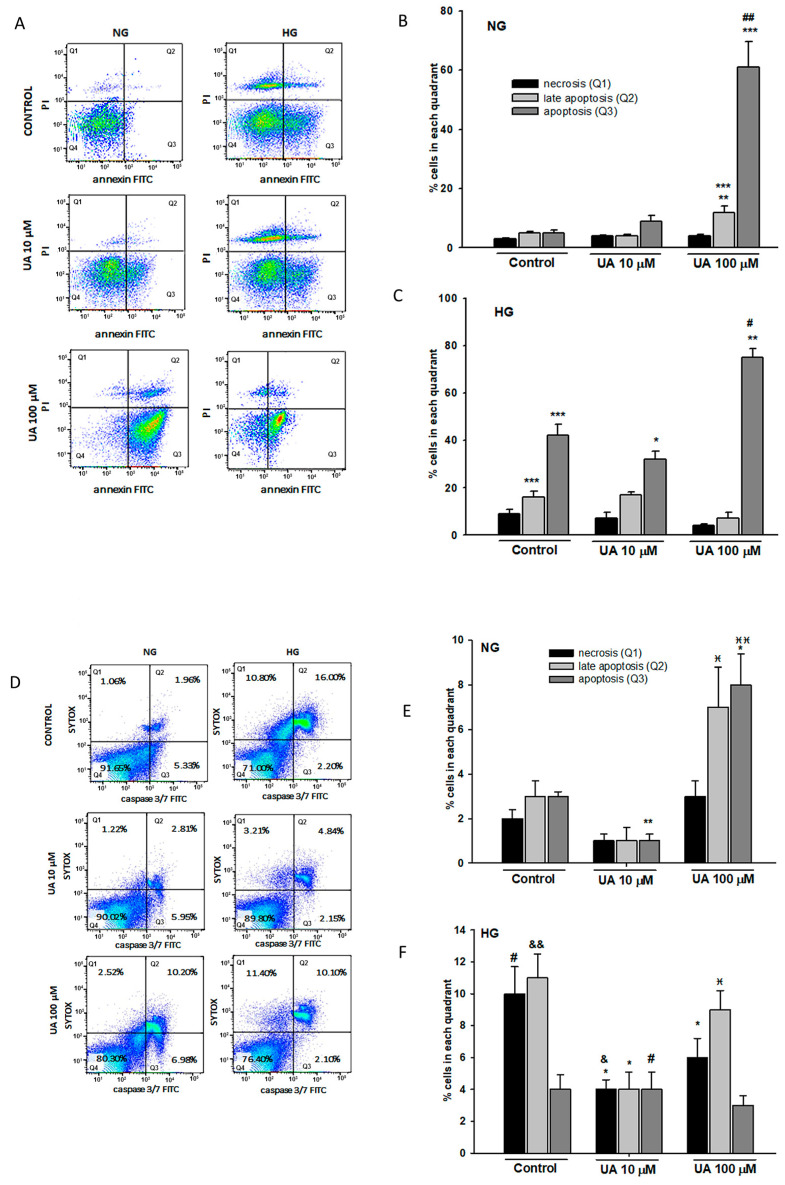
100 µM UA induced podocyte apoptosis. Podocytes cultured for 7 d in normal (5.5 mM, NG) or high (25 mM, HG) glucose were treated with 10 µM and 100 µM UA for 24 h. Representative flow cytometry images of Annexin V/Propidium Iodide (PI) double-staining (**A**) and quantitative analysis of podocyte apoptosis (**B**,**C**). Representative flow cytometry images of activated caspase-3/7 (**D**) quantitative analysis of activated caspase-3/7 detection (**E**,**F**). Sytox Green nucleic acid stain was used as an indicator of necrotic cells. ANOVA and the Student’s *t*-tests were used to calculate *p* values. For (**B**,**C**) * *p* < 0.05 vs. HG Control, # *p* < 0.01 vs. HG Control, ** *p* < 0.01 vs. respective 10 µM UA, ## *p* < 0.001 vs. NG- 10 µM UA, *** *p* < 0.001 vs. NG Control. For (**E**,**F**) * *p* < 0.05 vs. Control, ** *p* < 0.01 vs. Control, ^
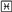
^
*p* < 0.05 vs. 10 µM UA, ^
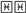
^
*p* < 0.01 vs. 10 µM UA, # *p* < 0.05 vs. respective NG group, & *p* < 0.01 vs. respective NG 10 µM UA, && *p* < 0.005 vs. NG Control. The values are mean ± SEM (*n* = 4 for (**B**,**C**), *n* = 3 for (**E**,**F**)).

**Figure 8 cells-11-02471-f008:**
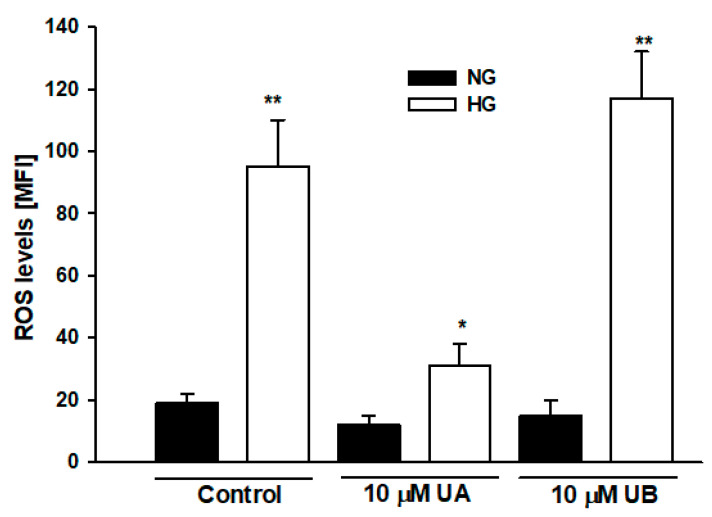
Urolithin A but not UB reduced reactive oxygen species (ROS) levels in podocytes exposed to high glucose. Podocytes cultured for 7 d in normal (5.5 mM, NG) or high (25 mM, HG) glucose were incubated for 24 h with 10 µM UA or 10 µM UB. ROS were detected by flow cytometry using CellROX Green reagent. Results show mean ± SEM. Mann–Whitney test was used to calculate *p* values. * *p* < 0.05 vs. HG Control, ** *p* < 0.01 vs. NG Control (*n* = 4). MFI: mean fluorescence intensity.

**Figure 9 cells-11-02471-f009:**
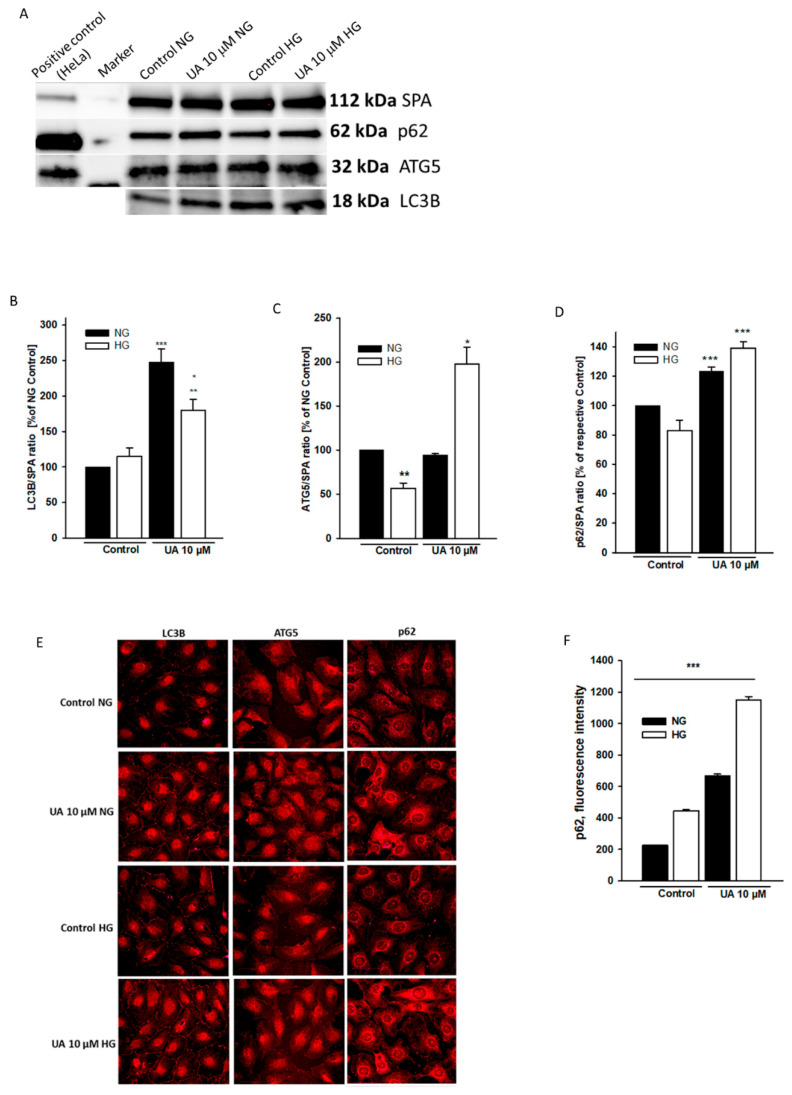
Expression of autophagic markers was upregulated by UA. Podocytes cultured for 7 d in normal (5.5 mM, NG) or high (25 mM, HG) glucose were incubated for 24 h with 10 µM UA. Immunofluorescence and Western blot analysis were performed to examine the expression patterns of autophagic markers LC3B, ATG-5, and p62. A representative immunoblot (**A**). Quantitative densitometric analysis was used to determine the ratios of LC3B (**B**), ATG-5 (**C**), and p62 (**D**) to Alpha 1 Sodium Potassium ATPase (SPA). Results of experiments performed in duplicate show mean ± SEM (*n* = 3 for ATG5 and p62, *n* = 4 for LC3B). The Student’s *t*-test was used to calculate *p* values. For LC3B (**B**) * *p* < 0.05 vs. HG Control, ** *p* < 0.01 vs. NG Control, *** *p* < 0.001 vs. NG Control. For ATG-5 (**C**) * *p* < 0.01 vs. NG-UA and vs. NG Control, ** *p* < 0.001 vs. NG Control and vs. HG-UA. For p62 (**D**) *** *p* < 0.001 vs. respective Control. Representative confocal microscopy images of immunofluorescent staining against LC3B, ATG-5, and p62 (**E**). Quantification of p62 expression (**F**). 500 cells were analyzed per experiment (*n* = 3), *** *p* < 0.001 NG vs. HG and UA vs. Control.

**Figure 10 cells-11-02471-f010:**
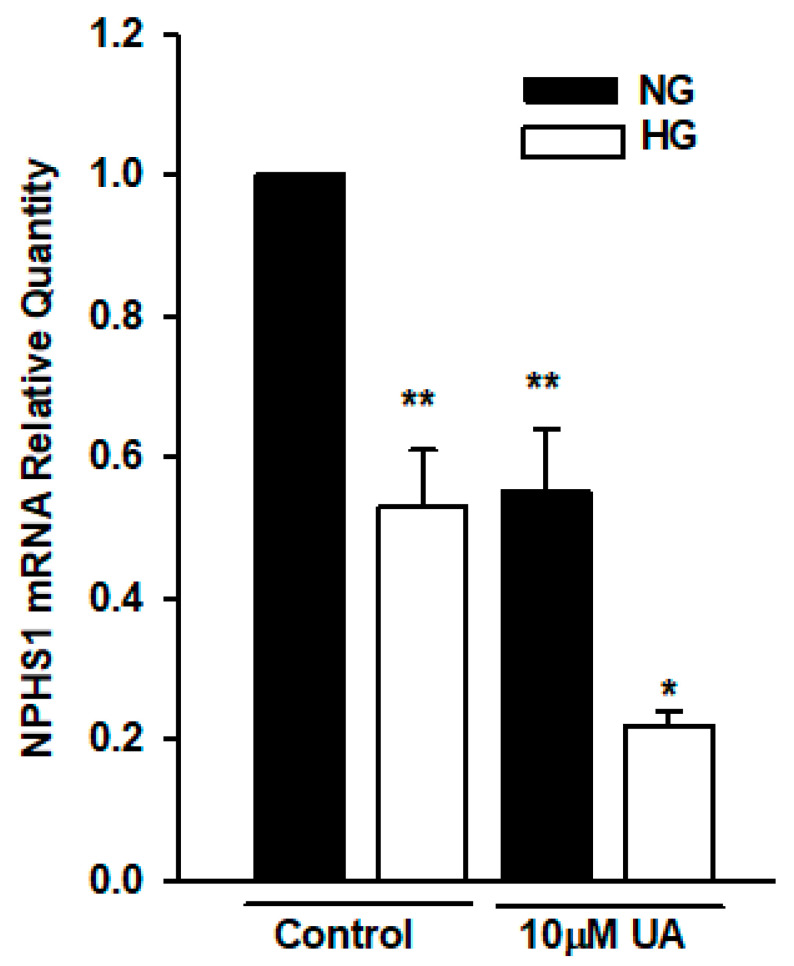
High glucose and UA downregulated the expression of NPHS1 mRNA. Podocytes cultured for 7 d in NG or HG media were incubated for 24 h with 10 µM UA, and total RNA was isolated and analyzed by quantitative RT-PCR. Relative levels of NPHS1 mRNA expression were normalized to β-actin. Each experiment was performed in triplicate, and results are shown as mean ± SEM. The Student’s *t*-test was used to calculate *p* values. * *p* < 0.05 vs. Control HG, ** *p* < 0.01 vs. Control NG (*n* = 3).

**Figure 11 cells-11-02471-f011:**
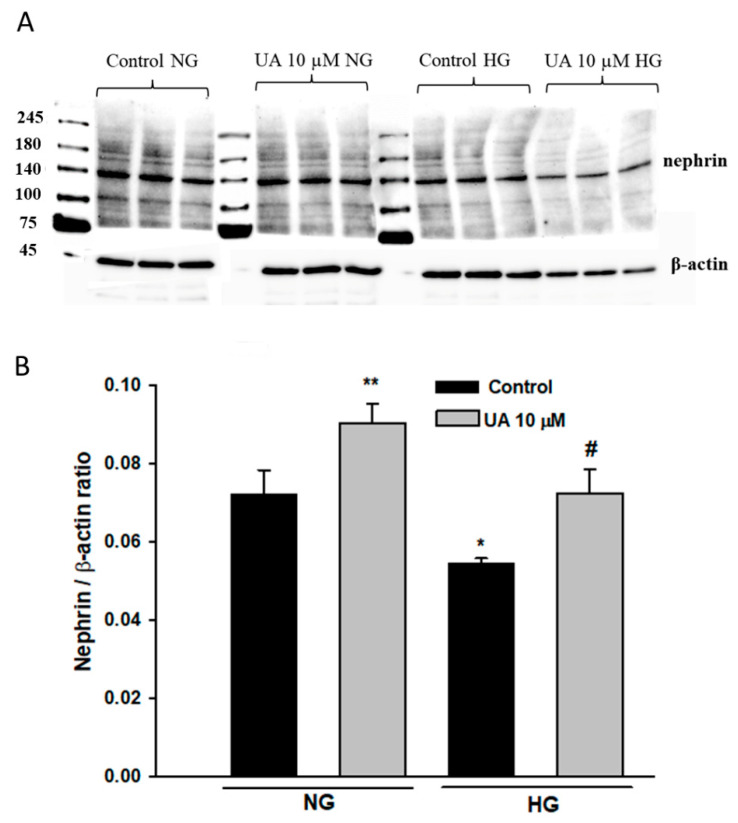
Urolithin A increased total nephrin expression in podocytes. 30-µg protein samples from total cell lysates were subjected to Western blot analysis followed by quantitative densitometric analysis. Nephrin expression in mouse kidney cortex homogenate served as a positive control. A representative immunoblot (**A**). Experiments were performed in triplicate. Results of quantitative densitometric analysis corrected for β-actin show mean ± SEM (**B**). The Student’s *t*-test was used to calculate *p* values. * *p* < 0.05, ** *p* < 0.01 vs. Control NG, # *p* < 0.05 vs. Control HG (*n* = 3–5).

**Figure 12 cells-11-02471-f012:**
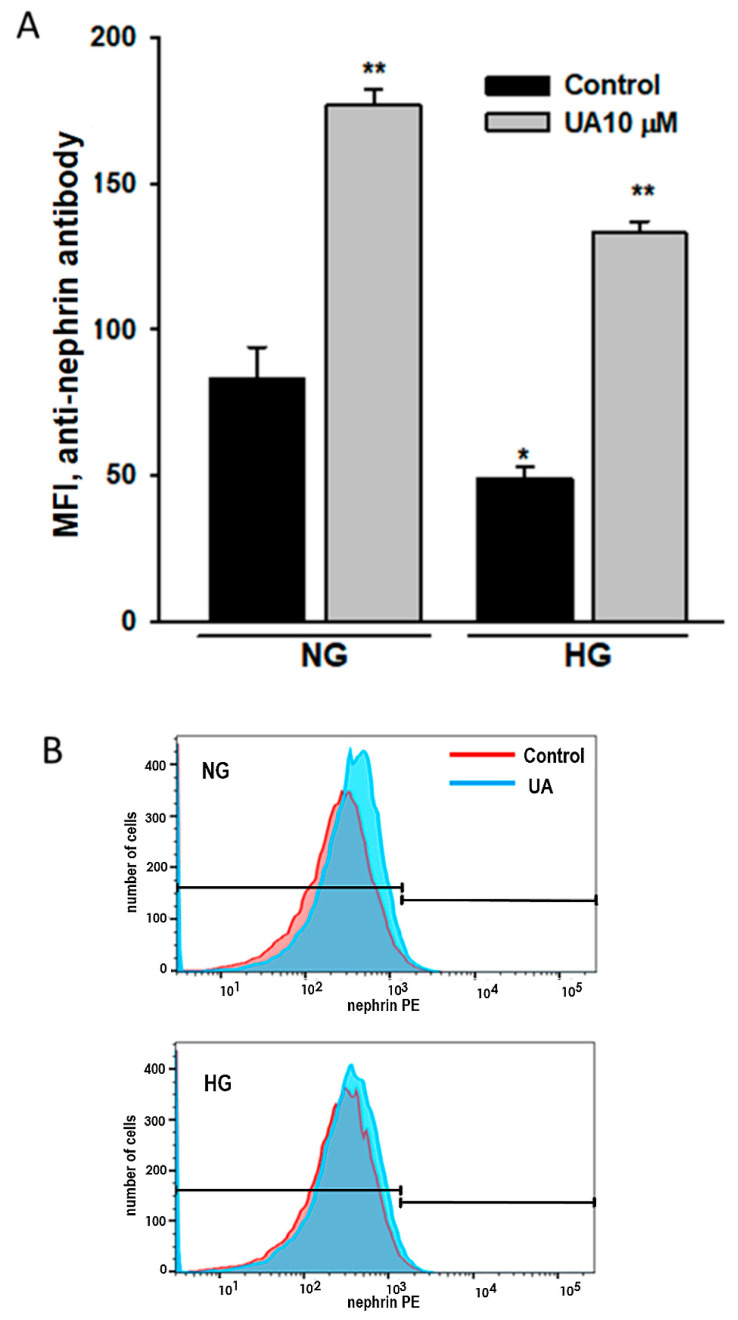
Urolithin A increased nephrin expression at the podocyte surface. Podocytes cultured for 7 d in normal (5.5 mM, NG) or high (25 mM, HG) glucose were incubated for 24 h with 10 µM UA, stained with phycoerythrin-conjugated antibody against the extracellular nephrin domain and analyzed by flow cytometry. Quantitative analysis of UA and UB effects (**A**) and representative histogram showing the effect of UA (**B**) on extracellular nephrin expression. Results show mean ± SEM. Mann–Whitney test was used to calculate *p* values. * *p* < 0.05 vs. NG Control, ** *p* < 0.001 vs. respective Control (*n* = 5). MFI: mean fluorescence intensity.

**Figure 13 cells-11-02471-f013:**
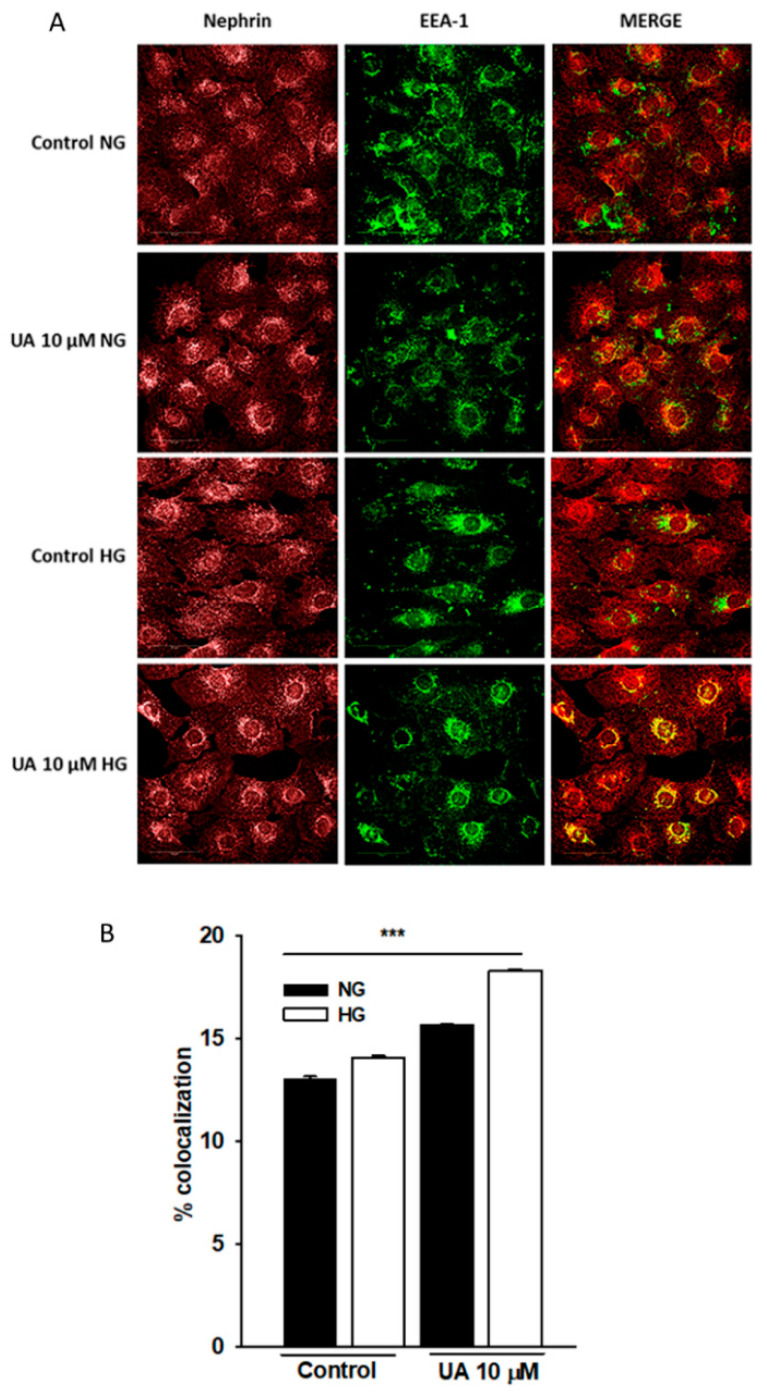
Upon treatment with UA, nephrin co-localization with early endosomal marker EEA1 was increased. Podocytes cultured for 7 d in normal (5.5 mM, NG) or high (25 mM, HG) glucose were incubated for 24 h with 10 µM UA. Representative confocal microscopy images of double staining against nephrin and EEA-1 (**A**). Quantification of nephrin and EEA-1 co-localization (**B**) Experiments were performed in duplicate and 500 cells were analyzed per experiment (*n* = 3). ANOVA test was used to calculate *p* values. *** *p* < 0.001.

## Data Availability

Not applicable.

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
