# Peer review of "Urolithins Modulate the Viability, Autophagy, Apoptosis, and Nephrin Turnover in Podocytes Exposed to High Glucose"

_cells, 2022, doi:10.3390/cells11162471_

Round 1

Reviewer 1 Report

Title Urolithins modulate the viability, autophagy, apoptosis and nephrin turnover in podocytes.

The authors describe the effects of UA and UB in low and high glucose concentrations on nephrin expression and podocyte morphology and viability.  The authors describe the background of their proposed studies and the findings well.

I have only minor considerations:

Line 99:  How low did the two glucose concentrations drop during the 7 day culture?  Was the culture media changed over the 7 days of culture?

Figure 10: For the Y axis legend, use: NPHS1 mRNA Relative Quantity

Figure 12A: rename Y axis legend for clarity: MFI anti-nephrin antibody

Author Response

Reviewer 1

We thank the Reviewer for the positive comments and careful review which helped improve the manuscript

Line 99:  How low did the two glucose concentrations drop during the 7 day culture?  Was the culture media changed over the 7 days of culture?

. We apologize for forgetting to mention this detail. We have now added the information  that the culture media were changed every 2 days.

We have established our high-glucose model almost 20 years ago  (BL  et al, Kidney Int 2004) and at the time, we have daily monitored changes in glucose level in NG (5.6 mM) and HG (30 mM) media which we used formerly. We used the results to establish the protocol for long-term exposition of podocytes to glucose. Our unpublished results are presented in the table below:

Day 0

Day 1

Day 2

Day 3

Day 4

NG

5.64± 0.2 (n=5)

5.5± 0.2 (n=5)

5.6± 0.1 (n=3)

5.3± 0.7 (n=3)

3.8± 0.16 (n=4)

HG

29.86± 0.8 (n=5)

29.8± 0.1 (n=5)

30.2± 0.4 (n=3)

27.5± 0.85 (n=3)

25.45± 0.94 (n=4)

Figure 10: For the Y axis legend, use: NPHS1 mRNA Relative Quantity

Figure 12A: rename Y axis legend for clarity: MFI anti-nephrin antibody

We agree that the axis legends were not clear. Following the Reviewer’s comments,  we now have re-named the axes.

Reviewer 2 Report

In this study, the authors report that in cultures of immortalized mouse podocytes, urolithin A (UA) improves cell viability and reduces apoptosis. In addition, high glucose-enhanced ROS formation is blocked by UA. The authors suggest that this effect of UA is due to increased nephrin protein expression, but also increased internalization. It is suggested that UA could serve as a potential therapeutic to prevent diabetic podocytopathy.

Major points:

1-In Fig. 9,A, the autopaghic markers are shown. No convincing differences between the lanes are seen. Why did the authors stain SPA as a control, while other Western blots show b-actin? The authors should stay consistent. In the original images of Fig. 9, SPA shows changes upon HG and UA and this makes an evaluation difficult. Also, the blots of LC3B show that the antibody is very unspecific and stains many bands in addition to LC3B. Therefore, immunofluorescence staining and quantification of the staining, as shown in Fig. 9E, is not useful.

2-In Fig. 10, the authors show that nephrin mRNA is downregulated by UA, while in Fig. 11 and 12, nephrin protein and surface localization was enhanced. The internalization of nephrin, shown by colocalization of nephrin with the early endosomal marker EEA1 was also enhanced. These data together make no sense. Also, Western blot data in Fig. 11A are of unacceptable poor quality and need to be improved. Notably, the supplied original blots for nephrin are too variable and do not show consistent results. The last set of data shows triplicates, which, however, show no increase by UA, but rather a decrease.  Since the MW marker sizes were not included, the size of the stained band can not be judged. It is highly recommended to show such triplicate blots in the main manuscript. Show the whole gel including molecular weight markers.

Most importantly, nephrin should run at approximately 200kDa, as it is a highly glycosylated protein, even if the calculated size is at 134 kDa. See also Ahola et al., 1999, AJP; Schiwetz et al., 2004, Kidney Int. Still, several bands could be seen due to spliced forms and fragments. Therefore, again, it is most important to show the whole gel range. Obviously, the antibody is not validated sufficiently, which then also questions the flow cytometry data in Fig. 12 and the immunofluorescence data in Fig. 13. Have stainings with only secondary antibody been done?

3-The authors here suggest that nephrin is the critical protein that protects the podocytes from apoptosis. However, this link is not proven. Is a downregulation of nephrin, for example by siRNA, causing apoptosis? ROS? Or are the two events (apoptosis and nephrin) independent of each other?

Author Response

Reviewer 2

We thank the Reviewer for the valuable comments and efforts towards improving our manuscript. In the following, we address the concerns of Reviewer.

1-In Fig. 9,A, the autopaghic markers are shown. No convincing differences between the lanes are seen. Why did the authors stain SPA as a control, while other Western blots show b-actin? The authors should stay consistent.

Thank you for your comment. It is right that one control protein should be used in all Western blots and we typically used b-actin.  However, according to the manufacturer, the mass of b-actin oscillates between 41-45 kDa, while the size of tested by us autophagy markers was  32 and 62 kDa which made it very difficult to stain together in one gel. That is why we decided to use SPA that runs higher than 100 kDa and could be better separated.  In some other publications, different reference proteins have also been used (Cieslik P et al, Int J Mol Sci 2021 , Park J-M et al, Sci Rep 2015)

In the original images of Fig. 9, SPA shows changes upon HG and UA and this makes an evaluation difficult.

We fully agree with the Reviewer that quality  of WB images for LC3B suggests changes in the reference protein.  For that reason, we have repeated the experiments 4 times and our autophagy evaluation included also other autophagy  markers. On the other hand , SPA images included in the images of ATG5 and p62 indicate that SPA did not change upon treatments . Supposedly, visual differences in bands could be due to uneven protein load . Yet, calculated LC3B to SPA ratios appeared to be within a narrow range of values, yielding consistent results . However, we thought it could be more convincing if we repeated the analysis once again. Unfortunately, there was no time enough to propagate cells and perform another experiment.

Also, the blots of LC3B show that the antibody is very unspecific and stains many bands in addition to LC3B. Therefore, immunofluorescence staining and quantification of the staining, as shown in Fig. 9E, is not useful

Thank you for your comment. The ab51520 antibody to LC3B has been shown  to react with  human (Abcam) as well as with mouse antigen (https://www.citeab.com/antibodies/741829-ab51520-anti-lc3b-antibody). We have used 4-20% gradient gel and multiple bands could result from high (30 µg) protein load. However, since we have stained also ATG5 and p62 at the same gel, we used relatively high lysate concentration so that all examined proteins could be well detected. Presumably, for LC3B it was excess but otherwise, other proteins would stain weak. According to the manufacturer’s specification, we used 18 kDa band for calculations.

In all IF analyzes, non-specific staining was controlled by replacing primary antibody with blocking solution alone which was followed by incubation with secondary antibody. Quantification of IF staining was done for p62 only.

Nevertheless, we will test new LC3B antibodies for our next experiments to avoid confusion.

In Fig. 10, the authors show that nephrin mRNA is downregulated by UA, while in Fig. 11 and 12, nephrin protein and surface localization was enhanced. The internalization of nephrin, shown by colocalization of nephrin with the early endosomal marker EEA1 was also enhanced. These data together make no sense.

We agree with the Reviewer that UA-dependent discrepancy between nephrin mRNA and protein expression seems to be confusing and our first feeling was just the same as the Reviewer’s. However,  repeated analyzes showed similar results and we had to accept it. Moreover, observed by us HG-induced downregulation of NPHS1 and protein is consistent with previously published data which confirms that our results are reliable. On the other hand, negative correlation between mRNA and protein is not unusual phenomenon (C.P. Moritz et al, https://doi.org/10.1111/jnc.14664, A. Koussounadis et al , https://doi.org/10.1038/srep10775, D. Wang doi: 10.1016/j.compbiolchem.2008.07.014) and is frequently attributed to different regulation between transcript and protein product. In our  (speculative) explanation of the discrepancy  included in the Discussion paragraph, we have pointed out that urolithins have been shown to affect  both transcription and translation which makes it possible that urolithin-dependent changes occurred also in our experimental model. In Discussion we also have proposed that enhanced endosomal trafficking could be due to the increased nephrin expression at the cell surface.

Also, Western blot data in Fig. 11A are of unacceptable poor quality and need to be improved.

According to the Revewer’s suggestion, we have replaced  Fig 11A with the full blot showing  triplicates and markers

Notably, the supplied original blots for nephrin are too variable and do not show consistent results. The last set of data shows triplicates, which, however, show no increase by UA, but rather a decrease. 

We agree with your concerns  that the differences between band intensities may look confusing. Indeed, the last three nephrin bands stained weaker than other bands, however this applies also to the b-actin bands. We have checked all calculated nephrin/actin intensity ratios from all experiments and results indicate UA enhanced total nephrin expression.

Since the MW marker sizes were not included, the size of the stained band can not be judged. It is highly recommended to show such triplicate blots in the main manuscript. Show the whole gel including molecular weight markers.

Most importantly, nephrin should run at approximately 200kDa, as it is a highly glycosylated protein, even if the calculated size is at 134 kDa. See also Ahola et al., 1999, AJP; Schiwetz et al., 2004, Kidney Int. Still, several bands could be seen due to spliced forms and fragments. Therefore, again, it is most important to show the whole gel range.

We fully agree with the Reviewer and following the suggestions, we have carefully checked again the mass of the product. We also have enhanced the image of the colorimetric  marker blot to make it clearly visible. With 30 µg protein loaded to 4-20%  gel, strong bands were close to 140 kDa

 Additionally, we have again compared podocyte lysate with kidney cortex homogenate. Indeed , in the kidney nephrin was about 180 kDa, as the Reviewer suggested. Below please find corresponding images We suppose that the degree of nephrin glycosylation could be different in our cultured immortalized cell line and in native podocytes from the kidney cortex. We are also aware that in publication by  D. Schiwek nephrin from similar to our model  murine podocytes stained at 180 kDa. However, as indicated in our next comment, in various publications  determined by WB nephrin masses vary between 134 and 200kDa.

We used the 3-color prestained protein marker (10-245 kDa, Blirt, Poland)

Obviously, the antibody is not validated sufficiently, which then also questions the flow cytometry data in Fig. 12 and the immunofluorescence data in Fig. 13.

For surface staining and for IF we used sc-376522 antibody (Santa Cruz) directed against the extracellular epitope of nephrin. The antibody was used by several other authors (e.g. C.J. Cooper et al doi: 10.1371/journal.pone.0203905, ·  DOI: 10.1152/ajprenal.00279.2020 ) ,  and validated by O. Bucur et al doi: 10.1038/s41596-020-0300-1)

For WB the ab216341 antibody was used. The ab216341antibody was also applied by other authors for IF (S. Clotet-Freixas et al https://doi.org/10.1681/ASN.2020030286, Y.Zang https://doi.org/10.1007/s11033-021-06204-4,) as well as for WB (L. Shi et al10.1080/0886022X.2022.2063744, Y. Lin et al https://doi.org/10.1155/2020/5803192, X. Yang et alhttps://doi.org/10.1590/fst.512210., L. Zhang et al,  doi: 10.1038/s41598-020-58781-2 ). Interestingly, in papers by Shi , by Zhang and by Lin nephrin was shown at 135 and at 136 kDa, while in paper by Yang it was 200kDa

Have stainings with only secondary antibody been done?

Routinely, all in sets for  FC blank controls for unspecific secondary antibody fluorescence were included and resulting MFI values were subtracted from the MFI for tested samples . Similarly, in all IF stainings  nonspecific controls were included. To clarify the procedure, in the Methods paragraph we have completed the sentence by adding the underlined text  Non-specific staining was controlled by replacing primary antibody with blocking solution alone which was followed by incubation with secondary antibody

Below  representative images of our control stainings are shown. Antibody to nephrin was already PE-conjugated but we have used blocking solution and isotype control serum as a control.  Quantitative analysis was performed by subtracting  fluorescence of the NS Control from fluorescence of tested cells.

The authors here suggest that nephrin is the critical protein that protects the podocytes from apoptosis. However, this link is not proven. Is a downregulation of nephrin, for example by siRNA, causing apoptosis? ROS? Or are the two events (apoptosis and nephrin) independent of each other?

We are grateful for the comment. We apologize for presenting  our results in a confusing way, leading to unintended by us suggestion that apoptosis was linked to nephrin expression. It is well documented that mutations and other disturbances  in nephrin expression cause podocyte impairment and loss,  however not via apoptotic mechanisms (Done et al, KI 2008).  The general goal of the present study was to examine the effects of urolithins on the functions podocytes, and particularly, in conditions mimicking diabetes. Based on the data from the studies on other cell types, we  first investigated the effects on viability and associated mechanisms. Since the results suggested that activity of urolithins could be beneficial to podocytes our next step was to examine the effects of urolithins on nephrin expression. We have focused just on nephrin because normal nephrin expression is particularly associated with proper function of  SD and podocytes.  To avoid misinterpretations, we have re-written the corresponding part of Discussion.

Reviewer 3 Report

The current report supports the potential beneficial effects of urolithins on renal function, further providing some mechanistic insights during conditions of hyperglycemia. The report is indeed of interest and provides diverse potential mechanisms that may be involved in the protective effects of urolithins. The obvious weakness of the study is because it mainly involves in vitro experiments, mostly cell viability assays. Thus, in vivo models could have solidified presented evidence, especially clarifying some of data showing discrepancies. However, please also consider the below comments to enhance quality of presented evidence:

Specific comments

It is better to already indicate on the title that you are looking at podocytes exposed to “High glucose”/ “hyperglycemia” conditions

Notably, cite relevance for controls for high glucose, including those accounting for osmotic stress?

In fact, the DMSO concentration reflecting 0.1% is very high, could you please provide citations indicating non-toxicity, or prior preliminary assays indicating that this dose/concentration was not toxic.

What was considered significant (p-value)? Indicate this under statistical analysis section.

Importantly, under each figure caption, which statistical analysis test was used, considering that you have mentioned both Mann-Whitney or Student’s t-tests.

The use of high glucose is not consistent, or rather check all abbreviations for consistency.

Under each figure caption, please indicate both biological and technical repeat number, to remain consistent. As figure one shows some experiments were done in duplicate (perhaps this could also indicate not seeing significance?).

Why not normalize Bcl-2 Bax? Before any actin or SPA?

The text/statements saying “antioxidant activity” within the caption or result section sating “improves oxidative stress induced by high glucose in podocytes” are incorrect. It is well established that oxidative stress encompasses both ROS and antioxidants, and in this case only ROS was measured. So rather be precise and discuss ROS not ROS

It could have provided more detail if result of high dose (100 um) vs. low dose (10 uM) in terms of understanding how urolithins modulate autophagy. Because the result will indicate whether urolithins promote or hinder this vital mechanistic process

To reduce the number of non-essential information, some figures, especially those reporting on cell viability can be rather combined.

Consistent with above comments, the statements, within the discussion, mentioning “upregulated autophagic flux and inhibited oxidative stress.” Eve “antioxidant” activity of urolithins should be amended or removed

Comment/discuss the varied therapeutic effects of UA and UB, within the discussed experimental model

Indicate limitations of the current study, considering its an in vitro experimental model?

Author Response

Reviewer 3

It is better to already indicate on the title that you are looking at podocytes exposed to “High glucose”/ “hyperglycemia” conditions

Thank you for your comment. Initially, we omitted “high glucose” because during experiments it appeared that also in NG conditions urolithins induced some changes. However indeed, our principal intention was to examine whether urolithins could improve high glucose-induced damage in podocytes. According to your suggestion, we have changed the title.

Notably, cite relevance for controls for high glucose, including those accounting for osmotic stress?

In our previous studies, (e.g. Lewko et al, Exp Diabetes Res 2011) to highlight the effects of glucose alone, we have used mannitol as osmotic control. However now we examined the responses of podocytes exposed to the milieu mimicking diabetes/hyperglycemia (ie high plasma glucose that is accompanied by increased osmolarity). In this context we considered separating glucose from osmotic effects to be unreasonable.  Therefore, in this study podocytes cultured in normal glucose (corresponding to non-diabetic) served as controls for HG (corresponding to diabetic).

In fact, the DMSO concentration reflecting 0.1% is very high, could you please provide citations indicating non-toxicity, or prior preliminary assays indicating that this dose/concentration was not toxic.

Prior to the experiments, we have checked the effects of DMSO concentration on podocyte viability . Our unpublished results are presented below

DMSO 0%

DMSO 0.01%

DMSO 0,05%

 DMSO 0.1%

DMSO 0.25%

Viability 100% n=5

99± 6% (n=5)

102± 8% (n=4)

93± 7%(n=4)

73± 5% (n=4)

Additionally, below please find citations

DMSO Concentrations up to 1% are Safe to be Used in the Zebrafish Embryo Developmental Toxicity Assay - PubMed (nih.gov) https://www.ncbi.nlm.nih.gov/pmc/articles/PMC8915880/

The ShGlomAssay Combines High-Throughput Drug Screening With Downstream Analyses and Reveals the Protective Role of Vitamin D3 and Calcipotriol on Podocytes https://www.frontiersin.org/articles/10.3389/fcell.2022.838086/full

Podocyte Injury Caused by Indoxyl Sulfate, a Uremic Toxin and Aryl-Hydrocarbon Receptor Ligand  https://www.ncbi.nlm.nih.gov/pmc/articles/PMC4171541/

What was considered significant (p-value)? Indicate this under statistical analysis section.

P<0.05 is considered significant. According to the Reviewer’s suggestion, we have indicated this in the text.

Importantly, under each figure caption, which statistical analysis test was used, considering that you have mentioned both Mann-Whitney or Student’s t-tests.

Thank you for the comment. Under each figure caption respective information has been included.

The use of high glucose is not consistent, or rather check all abbreviations for consistency.

Thank you again for careful checking the text. In this study, glucose concentration in HG media was 25mM . We have made respective corrections.   

Under each figure caption, please indicate both biological and technical repeat number, to remain consistent. As figure one shows some experiments were done in duplicate (perhaps this could also indicate not seeing significance?).

Thank you for the comment. We have added missing information under figure captions. Figures5,  6, and 7 present results from at least 3 independent experiments performed with no technical repeats.

Why not normalize Bcl-2 Bax? Before any actin or SPA?

As changes in Bcl-2 level reflect  both changes in  the cell pro-survival mode and autophagy, we determined Bcl-2 alone which was followed  by additional analysis of typical autophagic markers , as well as by FC test (Annexin V) for apoptosis. We fully agree that Bcl2/Bax would be appropriate and we are planning to present respective data in our next publication.

The text/statements saying “antioxidant activity” within the caption or result section sating “improves oxidative stress induced by high glucose in podocytes” are incorrect. It is well established that oxidative stress encompasses both ROS and antioxidants, and in this case only ROS was measured. So rather be precise and discuss ROS not ROS

The Reviewer is absolutely right, the term was used improperly. We now have introduced respective changes in the text to make it  more precise and relating to ROS only. To be more precise, we also have  changed the subtitle of the 3.3. paragraph.

It could have provided more detail if result of high dose (100 um) vs. low dose (10 uM) in terms of understanding how urolithins modulate autophagy. Because the result will indicate whether urolithins promote or hinder this vital mechanistic process

The Reviewer’s comment is in line with our plans. Unfortunately, we were not able to complete our present results with additional data.  Yet, considering unexpected discrepancy between p62 values and other autophagy markers, as well as rise in Bcl-2 at 100 µM urolithins we are going to explore urolithin- autophagy relationship more accurately and results will be presented in a separate manuscript.

To reduce the number of non-essential information, some figures, especially those reporting on cell viability can be rather combined.

Thank you for the comment. This was what we have considered already when composing the manuscript. However, Figure 3 demonstrates that viability is dose-dependent, while Figure  4 shows that at high glucose  UA and UB differ in their effects. We found it very difficult to combine all these data in one picture without losing clarity.

Consistent with above comments, the statements, within the discussion, mentioning “upregulated autophagic flux and inhibited oxidative stress.” Eve “antioxidant” activity of urolithins should be amended or removed

As mentioned in our previous response, we have corrected all statements on the antioxidant activity of urolithins, except citations.

Comment/discuss the varied therapeutic effects of UA and UB, within the discussed experimental model

Indicate limitations of the current study, considering its an in vitro experimental model?

Thank you for your helpful remarks. Following your suggestions,  we now have included new comments at the end of Discussion paragraph.

Round 2

Reviewer 2 Report

The authors have satisfactorily adressed the concerns

Reviewer 3 Report

The authors have adequately addressed my comments and the manuscript quality has been improved.